# Solving Inverse Problem for Multi-armed Bandits via Convex Optimization

## Abstract

We consider the inverse problem of multi-armed bandits (IMAB) that are widely used in neuroscience and psychology research for behavior modelling. We first show that the IMAB problem is not convex in general, but can be relaxed to a convex problem via variable transformation. Based on this result, we propose a two-step sequential heuristic for (approximately) solving the IMAB problem. We discuss a condition where our method provides global solution to the IMAB problem with certificate, as well as approximations to further save computing time. Numerical experiments indicate that our heuristic method is more robust than directly solving the IMAB problem via repeated local optimization, and can achieve the performance of Monte Carlo methods within a significantly decreased running time. We provide the implementation of our method based on `CVXPY`, which allows straightforward application by users not well versed in convex optimization.

## 1 Introduction

We consider optimization problems with the following structure:

$$
\begin{aligned}
\text{minimize} \quad & J = -\sum_{t=1}^{n} \ell(x(t), y(t)) \\
\text{subject to} \quad & x(t) = Ax(t-1) + Bu(t), \quad t = 1, \dots, n \\
& x(0) = 0,
\end{aligned}
\tag{1.1}
$$

where $x(t) \in \mathbf{R}^m$, $A, B \in \mathbf{R}^{m \times m}$ are the problem variables; $y(t), u(t) \in \mathbf{R}^m$ are the problem data. Specifically, we assume that the matrices $A$ and $B$ are diagonal and $y(t) \in \{e_1, \dots, e_m\}$ for all $t = 1, \dots, n$, where $e_i$ denotes the $i$th standard basis vector. The objective function has the form of

$$
J = -\sum_{t=1}^{n} \ell(x(t), y(t)) = -\sum_{t=1}^{n} \log \left( y(t)^T \left( \frac{\exp x(t)}{\sum_{i=1}^{m} \exp x_i(t)} \right) \right),
\tag{1.2}
$$

where $x_i(t)$ denotes the $i$th entry of the vector $x(t)$.

### 1.1 Problem interpretation and examples

The problem (1.1) can be interpreted as an inverse problem for multi-armed bandits (IMAB), which is widely used in behavioral neuroscience and cognitive science research for mathematical modeling of animal decision-making processes.

As a simple example, consider an agent facing repeatedly with a choice among $m$ different actions, with (potentially different) probabilities of obtaining a reward from each action. After the choice at time step $t-1$, the agent receives a *reward signal* $u(t) \in \mathbf{R}^m$ that depends on the selected action, *e.g.*,

$$
u_i(t) = \begin{cases} 1 & \text{if action } i \text{ was selected } and \text{ rewarded} \\ 0 & \text{otherwise,} \end{cases}
\tag{1.3}
$$

for $i = 1, \dots, m$. The (forward) learning problem of this $m$-armed bandit consists in maximizing the expected total reward over some time steps (Sutton & Barto, 2018, §2.1). For this purpose, one approach is to formulate

some *value function* (or really, vector) $x(t) \in \mathbf{R}^m$ for each time step $t$, and recursively updates it according to

$$x(t) = x(t-1) + \alpha(\beta u(t) - x(t-1)), \tag{1.4}$$

where the (given) parameters $\beta \in [0, \infty)$ can be considered as the sensitivity to the reward signal $u(t)$, and $\alpha \in [0, 1]$ is the learning rate of the value estimation error $(\beta u(t) - x(t-1))$. The initial value function is generally assumed to be $x(0) = 0$. Let $a(t) \in \{1, \ldots, m\}$ denote the agent's action at the $t$th time step, a widely considered decision model of the agent has the following form:

$$\mathbf{prob}(a(t) = i) = \frac{\exp x_i(t)}{\sum_{j=1}^m \exp x_j(t)}, \quad i = 1, \ldots, m, \quad t = 1, \ldots, n. \tag{1.5}$$

When the reward signal $u(t)$ has the form of (1.3), the decision model based on (1.4) and (1.5) is sometimes referred to as the *forgetting Q-learning* model (Ito & Doya, 2009). In contrast to the aforementioned forward problem with *given* $\alpha$ and $\beta$ to learn the value function $x(t)$, the corresponding inverse problem for this $m$-armed bandit is formulated as follows: Suppose we are given the agent's choice $a(t)$, and the corresponding reward signal $u(t)$ for $t = 1, \ldots, n$. The objective is to *find* a group of model parameters $\alpha$ and $\beta$ such that the likelihood of observing the given $a(t)$ and $u(t)$ (and hence the underlying value function $x(t)$) is maximized. Another name for this process which is more widely used by the behavioral neuroscience and cognitive science community is *"forgetting Q-learning model fitting"*. To formally write this inverse problem as (1.1), note that (1.4) can be written as

$$x(t) = (1 - \alpha)x(t-1) + \alpha\beta u(t) = Ax(t-1) + Bu(t)$$

with

$$A = I - \mathbf{diag}(\alpha, \ldots, \alpha) \in \mathbf{R}^{m \times m}, \quad B = \mathbf{diag}(\alpha\beta, \ldots, \alpha\beta) \in \mathbf{R}^{m \times m},$$

where $I$ is the identity matrix. Transforming the actions $a(t) \in \{1, \ldots, m\}$ into one-hot vector $y(t) \in \{e_1, \ldots, e_m\}$ as

$$y_i(t) = \begin{cases} 1 & a(t) = i \\ 0 & \text{otherwise,} \end{cases} \quad i = 1, \ldots, m, \tag{1.6}$$

the log-likelihood of observing $a(t)$ at time step $t$ is

$$\ell(x(t), y(t)) = \log\left(y(t)^T \left(\frac{\exp x(t)}{\sum_{i=1}^m \exp x_i(t)}\right)\right).$$

By adding up $-\ell(x(t), y(t))$ over all $t = 1, \ldots, n$ we obtain the objective (1.2). Put together, we have the following optimization problem:

$$
\begin{array}{ll}
\text{minimize} & -\sum_{t=1}^n \log(y(t)^T \exp x(t) / \sum_{i=1}^m \exp x_i(t)) \\
\text{subject to} & x(t) = (I - \mathbf{diag}(\alpha, \ldots, \alpha))x(t-1) + \mathbf{diag}(\alpha\beta, \ldots, \alpha\beta)u(t), \quad t = 1, \ldots, n \\
& x(0) = 0, \quad 0 \le \alpha \le 1, \quad \beta \ge 0
\end{array} \tag{1.7}
$$

over variable $\alpha$, $\beta$ with data $y(t)$, $u(t)$, as a specific instance of (1.1).

Several extensions of (1.7) have been proposed by different groups for different modelling contexts, but can nevertheless be formulated into the structure of (1.1). One useful extension is to incorporate different learning rate $\alpha$ and reward signal sensitivity $\beta$ for different actions (Hattori et al., 2019; 2023), *i.e.*,

$$A = I - \mathbf{diag}(\alpha_1, \ldots, \alpha_m), \quad B = \mathbf{diag}(\alpha_1\beta_1, \ldots, \alpha_m\beta_m),$$

where $\alpha_1, \ldots, \alpha_m \in [0, 1]$, $\beta_1, \ldots, \beta_m \in [0, \infty]$. Such an extension can be readily incorporated into (1.7). Another more complex extension assumes that there exist multiple *subreward signals* $u^{(1)}(t), \ldots, u^{(k)}(t) \in \mathbf{R}^m$ (Beron et al., 2022), corresponding to multiple *subvalue functions* $z^{(1)}(t), \ldots, z^{(k)}(t) \in \mathbf{R}^m$, which are updated individually according to (1.4), with respect to each subreward signal. The reward function $x(t)$ used in (1.5) is then a linear combination of $z^{(1)}(t), \ldots, z^{(k)}(t)$ under some *given* weight vector $w \in \mathbf{R}^k$, *i.e.*, $x(t) = w_1 z^{(1)}(t) + \cdots + w_k z^{(k)}(t)$. In many applications, the vector $w$ is assumed to be simply $\mathbf{1}$, *i.e.*, equal

weights are assigned to all subvalue functions. Putting these two extensions together, the problem (1.7) now becomes

$$
\begin{aligned}
\text{minimize} \quad & -\sum_{t=1}^{n} \log(y(t)^T \exp x(t) / \sum_{i=1}^{m} \exp x_i(t)) \\
\text{subject to} \quad & x(t) = Z(t)w \\
& z^{(i)}(t) = A^{(i)} z^{(i)}(t-1) + B^{(i)} u^{(i)}(t) \\
& A^{(i)} = I - \mathbf{diag}(\alpha_1^{(i)}, \ldots, \alpha_m^{(i)}), \quad B^{(i)} = \mathbf{diag}(\alpha_1^{(i)}\beta_1^{(i)}, \ldots, \alpha_m^{(i)}\beta_m^{(i)}) \\
& z^{(i)}(0) = 0, \quad 0 \le \alpha_j^{(i)} \le 1, \quad \beta_j^{(i)} \ge 0 \\
& i = 1, \ldots, k, \quad j = 1, \ldots, m, \quad t = 1, \ldots, n,
\end{aligned}
\tag{1.8}
$$

where $Z(t) = \begin{bmatrix} z^{(1)}(t) & \cdots & z^{(k)}(t) \end{bmatrix} \in \mathbf{R}^{m \times k}$. Note that the variables for (1.8) are $\alpha_1^{(i)}, \ldots, \alpha_m^{(i)}$, $\beta_1^{(i)}, \ldots, \beta_m^{(i)}$, $i = 1, \ldots, k$; the problem data are $y(t)$, $u^{(i)}(t)$ for $t = 1, \ldots, n$, $i = 1, \ldots, k$, and $w$. (The other variables are slack variables introduced to simplify the notation.)

## 1.2   Previous and related work

The IMAB problem (1.1) is not convex in general (as we will show in §2.1). Different methods have been applied to solve (1.1), which can be roughly categorized into two groups: local minimization methods and Monte Carlo methods. The implementation of these methods are supported by different domain specific languages (DSLs).

Local minimization methods (or *direct* methods) for (1.1) consist in finding locally optimal solution repeatedly from different initial points and selecting the one with the minimum objective value across the repetitions as the final solution (Hattori et al., 2019; Beron et al., 2022; Hattori et al., 2023). The major advantage of these methods is that they are very easy to implement and are supported by different DSLs, such as `SciPy` (Virtanen et al., 2020), `JAX` (Bradbury et al., 2018), *etc.* Hence, direct methods have become the most widely used class of methods for IMAB problems.

The general idea of Monte Carlo methods is that instead of directly searching for the optimal variables of (1.1), they try to estimate the posterior distribution $p(A, B \mid y(t), u(t), \ t = 1, \ldots, n)$ of the variables given the problem data and some predefined prior, by generating a large number of instances for $A$ and $B$ from the desired distribution. The users then select the group of variables by taking, *e.g.*, the mean of the distribution, as the solution of (1.1) (Hamaguchi et al., 2022). In practice, Monte Carlo methods have better performance in solving (1.1) than direct methods, since global information about the optimal point is incorporated, but the implementation based on the DSL `PyMC` (Abril-Pla et al., 2023) can be more complex and tricky.

## 1.3   Contribution

Both widely used direct methods and Monte Carlo methods for solving (1.1) have limitations (as we show empirically in §4). Results from direct methods strongly depend on the number of initial points and their location and thus can be far from global optimum due to bad luck (and the users can not evaluate the suboptimality of the obtained result). Monte Carlo methods are able to generate robust results close to global optimum, but are in general very slow since a large number of samples have to be drawn from the targeted distribution.

In this work, we introduce a novel method for solving (1.1) by transforming the original problem into a sequence of two problems. The first problem is a relaxed problem of (1.1) by introducing convex relaxations via variable transformation, which can be solved efficiently via convex optimization; then solving the second problem recovers the original variables from the transformed variables. Numerical experiments show that our method is able to achieve performance similar to that of Monte Carlo methods with a significantly reduced computing time. Besides, our approach provides information about the suboptimality of the obtained approximate solution to (1.1), or even certificate for global optimality under some circumstances, which can be valuable for computational science applications. We provide a simple implementation of our method based on the DSL `CVXPY` (Diamond & Boyd, 2016; Agrawal et al., 2018) for convex optimization at

https://anonymous.4open.science/r/cvx_imab-A8C0,

such that users can easily apply our method to their problems, without the background knowledge about convex optimization.

## 2 IMAB problem analysis and convexification

In this section, we provide some analysis on a specific instance of the IMAB problem (1.1) given by

$$
\begin{array}{ll}
\text{minimize} & J = -\sum_{t=1}^{n} \log(y(t)^T \exp x(t) / \sum_{i=1}^{m} \exp x_i(t)) \\
\text{subject to} & x(t) = Ax(t-1) + Bu(t) \\
& A = I - \mathbf{diag}(\alpha_1, \ldots, \alpha_m), \quad B = \mathbf{diag}(\alpha_1\beta_1, \ldots, \alpha_m\beta_m) \\
& x(0) = 0, \quad 0 \le \alpha_i \le 1, \quad \beta_i \ge 0 \\
& i = 1, \ldots, m, \quad t = 1, \ldots, n.
\end{array}
\tag{2.1}
$$

The problem (2.1) is a generalized version of (1.7), thus all subsequent results based on (2.1) also hold for (1.7). For the simplicity of notation, we will not provide analysis about the problem formulation (1.8); all corresponding results from (2.1) can be readily extended to (1.8) via basic convex analysis (see (Rockafellar, 1970, §5) and (Boyd & Vandenberghe, 2004, §3.2)).

### 2.1 Problem transformation and convexity analysis

We start by eliminating the recursive expression about $x(t)$ in (2.1). Considering the $i$th entry of the vectors $x(0), \ldots, x(n)$, we have

$$
\begin{aligned}
x_i(0) &= 0 \\
x_i(1) &= (1 - \alpha_i)x_i(0) + \alpha_i\beta_iu_i(1) = \alpha_i\beta_iu_i(1) \\
x_i(2) &= (1 - \alpha_i)x_i(1) + \alpha_i\beta_iu_i(2) = (1 - \alpha_i)\alpha_i\beta_iu_i(1) + \alpha_i\beta_iu_i(2) \\
x_i(3) &= (1 - \alpha_i)x_i(2) + \alpha_i\beta_iu_i(3) = (1 - \alpha_i)^2\alpha_i\beta_iu_i(1) + (1 - \alpha_i)\alpha_i\beta_iu_i(2) + \alpha_i\beta_iu_i(3) \\
&\vdots \\
x_i(n) &= (1 - \alpha_i)x_i(n-1) + \alpha_i\beta_iu_i(n) = \sum_{t=1}^{n} (1 - \alpha_i)^{n-t}\alpha_i\beta_iu_i(t).
\end{aligned}
$$

Hence, the value function $x(t)$ for any $t = 1, \ldots, n$ can be expressed as

$$
x(t) = \mathbf{diag}(F\tilde{U}(t)),
$$

where

$$
F = \begin{bmatrix}
\alpha_1\beta_1 & (1 - \alpha_1)^1\alpha_1\beta_1 & \cdots & (1 - \alpha_1)^{n-1}\alpha_1\beta_1 \\
\vdots & \vdots & \ddots & \vdots \\
\alpha_m\beta_m & (1 - \alpha_m)^1\alpha_m\beta_m & \cdots & (1 - \alpha_m)^{n-1}\alpha_m\beta_m
\end{bmatrix} \in \mathbf{R}^{m \times n}.
\tag{2.2}
$$

The matrices $\tilde{U}(t)$, $t = 1, \ldots, n$ are defined as

$$
\tilde{U}(t) = \begin{bmatrix} U(t) \\ 0 \end{bmatrix} \in \mathbf{R}^{n \times m}, \quad U(t) = \begin{bmatrix} u(t)^T \\ \vdots \\ u(1)^T \end{bmatrix} \in \mathbf{R}^{t \times m}.
\tag{2.3}
$$

The matrices $\tilde{U}(t)$ given by (2.3) can be interpreted as follows: For each $t = 1, \ldots, n$, the matrix $\tilde{U}(t)$ is formed by padding an $(n - t) \times m$ matrix with all entries zero to the end of the corresponding reward signal

matrix $U(t)$, such that $\tilde{U}(t) \in \mathbf{R}^{n \times m}$. Putting together, the problem (2.1) can be transformed to

$$
\begin{aligned}
\text{minimize} \quad & J = -\sum_{t=1}^{n} \log(y(t)^T \exp x(t) / \sum_{i=1}^{m} \exp x_i(t)) \\
\text{subject to} \quad & x(t) = \mathbf{diag}(F\tilde{U}(t)) \\
& F = \begin{bmatrix} \alpha_1 \beta_1 & (1-\alpha_1)^1 \alpha_1 \beta_1 & \cdots & (1-\alpha_1)^{n-1} \alpha_1 \beta_1 \\ \vdots & \vdots & \ddots & \vdots \\ \alpha_m \beta_m & (1-\alpha_m)^1 \alpha_m \beta_m & \cdots & (1-\alpha_m)^{n-1} \alpha_m \beta_m \end{bmatrix} \\
& 0 \le \alpha_i \le 1, \quad \beta_i \ge 0, \quad i = 1, \dots, m, \quad t = 1, \dots, n,
\end{aligned}
\tag{2.4}
$$

where the variables are $\alpha_1, \dots, \alpha_m$, $\beta_1, \dots, \beta_m$ (with slack variable $F$); the data are $y(t)$ and $\tilde{U}(t)$, $t = 1, \dots, n$, given by (1.6) and (2.3).

We can easily obtain the convexity of the problem (2.1) by analyzing the transformed problem (2.4). Noting the objective function $J$ in (2.4) can be written as

$$
J = -\sum_{t=1}^{n} \log \left( \frac{y(t)^T \exp x(t)}{\sum_{i=1}^{m} \exp x_i(t)} \right) = -\sum_{t=1}^{n} \left( y(t)^T x(t) - \log \sum_{i=1}^{m} \exp x_i(t) \right),
$$

it follows immediately that the function $J$ is convex with respect to $\alpha_i$, $\beta_i$, $i = 1, \dots, m$, if and only if $x(t)$ are affine of variables $\alpha_i$, $\beta_i$, $i = 1, \dots, m$ (Grant, 2004; Grant et al., 2006). Since $x(t) = \mathbf{diag}(F\tilde{U}(t))$ where the righthand side is a linear transformation of $F$, the previous requirement is equivalent to saying that the transformation

$$
f \colon (\alpha_1, \dots, \alpha_m, \beta_1, \dots, \beta_m) \mapsto F
\tag{2.5}
$$

is affine. However, this is violated as a result of the polynomial terms about $\alpha_i$ and $\beta_i$ in (2.2). Hence, we come to the conclusion that the problem (2.1) is *not* convex.

## 2.2 Convexification via relaxation

We now introduce a convex relaxation on (2.4). Let $\eta = (1 - \alpha_1, \dots, 1 - \alpha_m) \in \mathbf{R}^m$. By variable transformation we have an equivalent problem of (2.4) given by

$$
\begin{aligned}
\text{minimize} \quad & J = -\sum_{t=1}^{n} \log(y(t)^T \exp x(t) / \sum_{i=1}^{m} \exp x_i(t)) \\
\text{subject to} \quad & x(t) = \mathbf{diag}(G\tilde{U}(t)), \quad t = 1, \dots, n \\
& G = \begin{bmatrix} g_1^T \\ \vdots \\ g_m^T \end{bmatrix} = \begin{bmatrix} \tilde{g}_1 & \cdots & \tilde{g}_n \end{bmatrix} \\
& g_i \succeq 0, \quad i = 1, \dots, m \\
& \tilde{g}_{i+1} = \mathbf{diag}(\eta)\tilde{g}_i, \quad i = 1, \dots, n - 1 \\
& 0 \preceq \eta \preceq 1,
\end{aligned}
\tag{2.6}
$$

where the vectors $g_1, \dots, g_m \in \mathbf{R}^n$ and $\tilde{g}_1, \dots, \tilde{g}_n \in \mathbf{R}^m$ are the vectors from the rows and columns of the matrix $G \in \mathbf{R}^{m \times n}$, respectively. (The notation $\succeq$ means componentwise inequality between vectors.) Note that the variables of (2.6) are the matrix $G \in \mathbf{R}^{m \times n}$ and $\eta \in \mathbf{R}^m$.

The problem (2.6) is equivalent to (2.4) and hence is not convex, but can be easily convexified by relaxing the last two constraints, *i.e.*, replacing

$$
\tilde{g}_{i+1} = \mathbf{diag}(\eta)\tilde{g}_i, \quad i = 1, \dots, n - 1, \quad 0 \preceq \eta \preceq 1
\tag{2.7}
$$

with

$$
\tilde{g}_1 \succeq \cdots \succeq \tilde{g}_n.
\tag{2.8}
$$

This leads to the problem:

$$
\begin{aligned}
\text{minimize} \quad & J = -\sum_{t=1}^{n} \log(y(t)^T \exp x(t) / \sum_{i=1}^{m} \exp x_i(t)) \\
\text{subject to} \quad & x(t) = \mathbf{diag}(G\tilde{U}(t)), \quad t = 1, \ldots, n \\
& G = \begin{bmatrix} g_1^T \\ \vdots \\ g_m^T \end{bmatrix} = \begin{bmatrix} \tilde{g}_1 & \cdots & \tilde{g}_n \end{bmatrix} \\
& g_i \succeq 0, \quad i = 1, \ldots, m \\
& \tilde{g}_1 \succeq \cdots \succeq \tilde{g}_n.
\end{aligned}
\tag{2.9}
$$

Convexity of (2.9) follows from the analysis in §2.1 and that $x(t)$ are now simply linear transformations of $G$ given each $\tilde{U}(t)$, and all inequality constraints on $G$ are linear. Such a convex relaxation can be interpreted as follows: Constraints (2.7) require that entries of $G$ decay *exponentially* under factor $\eta$ along each row, whereas (2.8) only requires that the entries decay along each row, *but not necessarily have to be exponentially.*

Let $C$ and $D$ be the set of all feasible matrices $G$ in (2.6) and (2.9), respectively. We then have $C \subseteq D$, and it follows immediately that by solving (2.9), a lower bound to the optimal value of problem (2.6) (and hence (2.4)) is obtained.

## 3 Heuristic for IMAB approximate solution

Based on the analysis in the previous section, we propose the following heuristic for (approximately) solving the IMAB problem (2.1) (and can be readily adapted for different formulations of (1.1)):

---

**given** data $y(t)$, $\tilde{U}(t)$, $t = 1, \ldots, n$.

Step I. *Solve the relaxed problem (2.9) via convex optimization to obtain the optimal variable $G^\star$.*

Step II. *Find a group of variables $(\alpha_1^\star, \ldots, \alpha_m^\star, \ \beta_1^\star, \ldots, \beta_m^\star)$ that satisfies $0 \le \alpha_i^\star \le 1$, $\beta_i^\star \ge 0$, $i = 1, \ldots, m$, such that its image $F^\star$ under the transformation (2.5) is the "closest" to $G^\star$.*

---

To formally write the second step in our heuristic as an optimization problem, noting that the $i$th row of $F$ only depends on $\alpha_i$ and $\beta_i$, we have:

$$
\begin{aligned}
\text{minimize} \quad & L = \sum_{i=1}^{m} \phi(\tilde{f}(\alpha_i, \beta_i), g_i^\star) = \sum_{i=1}^{m} \|\tilde{f}(\alpha_i, \beta_i) - g_i^\star\|_2^2 \\
\text{subject to} \quad & 0 \le \alpha_i \le 1, \quad \beta_i \ge 0, \quad i = 1, \ldots, m,
\end{aligned}
\tag{3.1}
$$

where the transformation $\tilde{f}$ is given by

$$
\tilde{f}: (\alpha, \beta) \mapsto (\alpha\beta, \ (1-\alpha)^1 \alpha\beta, \ \cdots, \ (1-\alpha)^{n-1} \alpha\beta)
$$

(*cf.*, (2.2) and (2.5)), and the penalty function $\phi(u, v) = \|u - v\|_2^2$ is the least squares penalty function. The variables of (3.1) are $\alpha_i, \beta_i \in \mathbf{R}$, $i = 1, \ldots, m$; the data are the vectors $g_i^\star \in \mathbf{R}^n$, $i = 1, \ldots, m$, which are from the rows of the matrix $G^\star$.

### 3.1 Implementation

The step I problem (2.9) in our heuristic can be easily specified and solved by the DSL `CVXPY`. The step II problem (3.1), however, is *not* convex[1]. Here we provide an option of trying to solve (3.1) in practice, where global optimum can *sometimes* be achieved, depending on the problem data (*cf.*, §3.2).

---

[1] By selecting a different penalty function $\phi$, we can actually obtain a convex problem formulation for the second step of our heuristic. We provide some discussion about this option in §A, but we will not consider this approach subsequently in this paper, for the reason listed there.

From the structure of (3.1), we immediately recognize a lower bound of its objective: $L^{\mathrm{lb}} = 0$. Hence, every time we locally solve (3.1) from a random initial point, the gap $|L^{\mathrm{loc}} - L^{\mathrm{lb}}|$ between the achieved optimal value $L^{\mathrm{loc}}$ (which is an upper bound of (3.1)) and the lower bound $L^{\mathrm{lb}}$ provides us a global optimality measure. Suppose that we repeatedly optimize (3.1) $N$ times from different initial point, and let $L^{(k)}$, $\alpha_1^{(k)}, \ldots, \alpha_m^{(k)}$, $\beta_1^{(k)}, \ldots, \beta_m^{(k)}$ be the returned (locally) optimal value and optimal variables from the $k$th run, respectively. If the gap $|L^{(k)} - L^{\mathrm{lb}}| \le \epsilon$ for some $k$, where $\epsilon$ is some predefined very small number, we can conclude that the *global* minimum $L^\star = L^{(k)}$ and is achieved at $\alpha_1^{(k)}, \ldots, \alpha_m^{(k)}, \beta_1^{(k)}, \ldots, \beta_m^{(k)}$. Otherwise, we can take the best objective value and the corresponding variables across the $N$ repetitions as an (approximate) solution, *i.e.*,

$$L^\star = \min\{L^{(1)}, \ldots, L^{(N)}\}, \quad (\alpha_1^\star, \ldots, \alpha_m^\star, \beta_1^\star, \ldots, \beta_m^\star) = \operatorname{argmin}\{L^{(1)}, \ldots, L^{(N)}\}.$$

Intuitively, this procedure returns a (potentially approximate) solution to (2.1) that is *at least* as suboptimal as the direct methods (*cf.*, §4).

Put together, we have the full implementation of our heuristic for approximately solving (2.1):

---

**Algorithm 1** *Heuristic for approximately solving the IMAB problem.*

---

**given** data $y(t)$, $\tilde{U}(t)$, $t = 1, \ldots, n$, tolerance $\epsilon > 0$, repeats $N$.

Step I. *Solve the relaxed problem (2.9).*
$$G^\star := \operatorname*{argmin}_{G} -\sum_{t=1}^{n} \log\left( \frac{y(t)^T \operatorname{diag}(G\tilde{U}(t))}{\mathbf{1}^T \exp\operatorname{diag}(G\tilde{U}(t))} \right), \text{ subject to } g_1, \ldots, g_m \succeq 0, \tilde{g}_1 \succeq \cdots \succeq \tilde{g}_n.$$

Step II. *Solve the function fitting problem (3.1).*
    **repeat** for $k = 1, \ldots, N$
        Compute $L^{(k)}$, $\alpha_i^{(k)}$, $\beta_i^{(k)}$, $i = 1, \ldots, m$ by locally minimizing $\sum_{i=1}^{m} \|\tilde{f}(\alpha_i, \beta_i) - g_i^\star\|_2^2$ from random
        initial point, subject to $0 \le \alpha_i \le 1$, $\beta_i \ge 0$, $i = 1, \ldots, m$.
        **if** $|L^{(k)} - L^{\mathrm{lb}}| \le \epsilon$
            $L^\star := L^{(k)}$, $\alpha_i^\star := \alpha_i^{(k)}$, $\beta_i^\star := \beta_i^{(k)}$, $i = 1, \ldots, m$.
            **quit**.
    $L^\star := \min\{L^{(1)}, \ldots, L^{(N)}\}$.
    $(\alpha_1^\star, \ldots, \alpha_m^\star, \beta_1^\star, \ldots, \beta_m^\star) := \operatorname{argmin}\{L^{(1)}, \ldots, L^{(N)}\}$.

---

## 3.2 Condition for exact solution

There is a special case where we can conclude that the result from algorithm 1 is actually the *exact* (or global optimal) solution to (2.1).

Consider the case where algorithm 1 has an early quit during step II as $|L^{(k)} - L^{\mathrm{lb}}| \le \epsilon$, and returns $L^\star = L^{(k)}$, $\alpha_i^\star = \alpha_i^{(k)}$, $\beta_i^\star = \beta_i^{(k)}$, $i = 1, \ldots, m$. We then have

$$\begin{bmatrix} \alpha_1^\star\beta_1^\star & (1-\alpha_1^\star)^1\alpha_1^\star\beta_1^\star & \cdots & (1-\alpha_1^\star)^{n-1}\alpha_1^\star\beta_1^\star \\ \vdots & \vdots & \ddots & \vdots \\ \alpha_m^\star\beta_m^\star & (1-\alpha_m^\star)^1\alpha_m^\star\beta_m^\star & \cdots & (1-\alpha_m^\star)^{n-1}\alpha_m^\star\beta_m^\star \end{bmatrix} = G^\star \tag{3.2}$$

within tolerance $\epsilon$. By performing elementwise division between each neighboring columns, we conclude that

$$\tilde{g}_{i+1} = \operatorname{diag}(1 - \alpha_1^\star, \ldots, 1 - \alpha_m^\star)\tilde{g}_i, \quad i = 1, \ldots, n-1, \tag{3.3}$$

where $\tilde{g}_i$ are the columns of $G^\star$, *i.e.*, the exponential decay constraint (2.8) is satisfied with $\eta = (1 - \alpha_1^\star, \ldots, 1 - \alpha_m^\star)$. Recalling that in §2.2 we have $C \subseteq D$ between the feasible sets for the matrices $F \in C$ and $G \in D$, the condition (3.3) says that we also have $G^\star \in C$, *i.e.*, the problems (2.6) and (2.9) share the same optimal point. Hence, in this case, we can conclude that the heuristic solution from algorithm 1 is actually the exact solution for (2.1).

Another interpretation for this exact solution condition is the following: By evaluating the objective $J$ of (2.1) with variables $\alpha_i^{(k)}$ and $\beta_i^{(k)}$ corresponding to $|L^{(k)} - L^{\mathrm{lb}}| \le \epsilon$, we obtain an upper bound $J^{\mathrm{ub}}$. Since

the variables $\alpha_i^{(k)}$ and $\beta_i^{(k)}$ satisfy the equality (3.2), and evaluating the objective of (2.9) with $G^\star$ provides a lower bound $J^{\mathrm{lb}}$ to the optimal value of (2.1), the upper bound $J^{\mathrm{ub}}$ and the lower bound $J^{\mathrm{lb}}$ must be equal. Hence, we have the (globally) optimal value $J^\star$ of (2.1) satisfies $J^{\mathrm{lb}} = J^\star = J^{\mathrm{ub}}$, and is achieved by $\alpha_1^{(k)}, \ldots, \alpha_m^{(k)}, \beta_1^{(k)}, \ldots, \beta_m^{(k)}$.

### 3.3 Truncated polynomial approximation

Solving the relaxed problem (2.9) can be slow for long horizon (*i.e.*, $n$ large) IMAB problems, since we need to find all the entries for a large matrix $G \in \mathbf{R}^{m \times n}$. To address this limitation, recall that the entries of each row in the matrix $F$ given by (2.2) decay exponentially, and hence, it is reasonable to approximate the matrix $F$ as $F = \begin{bmatrix} F_p & 0 \end{bmatrix}$, where

$$F_p = \begin{bmatrix} \alpha_1\beta_1 & (1-\alpha_1)^1\alpha_1\beta_1 & \cdots & (1-\alpha_1)^{p-1}\alpha_1\beta_1 \\ \vdots & \vdots & \ddots & \vdots \\ \alpha_m\beta_m & (1-\alpha_m)^1\alpha_m\beta_m & \cdots & (1-\alpha_m)^{p-1}\alpha_m\beta_m \end{bmatrix} \in \mathbf{R}^{m \times p},$$

*i.e.*, assuming that all the entries in $F$ after the $p$th column have decayed to be zero. Then the problem (2.1) reduces to

$$
\begin{aligned}
\text{minimize} \quad & J = -\sum_{t=1}^n \log(y(t)^T \exp x(t) / \sum_{i=1}^m \exp x_i(t)) \\
\text{subject to} \quad & x(t) = \mathbf{diag}(F_p \tilde{U}_p(t)) \\
& F_p = \begin{bmatrix} \alpha_1\beta_1 & (1-\alpha_1)^1\alpha_1\beta_1 & \cdots & (1-\alpha_1)^{p-1}\alpha_1\beta_1 \\ \vdots & \vdots & \ddots & \vdots \\ \alpha_m\beta_m & (1-\alpha_m)^1\alpha_m\beta_m & \cdots & (1-\alpha_m)^{p-1}\alpha_m\beta_m \end{bmatrix} \\
& 0 \le \alpha_i \le 1, \quad \beta_i \ge 0, \quad i = 1, \ldots, m, \quad t = 1, \ldots, n,
\end{aligned}
\tag{3.4}
$$

where the data $\tilde{U}_p(t)$ are simply the first $p$ rows of $\tilde{U}(t)$ given by (2.3). As a result, in (2.9), we only need to optimize over the matrix $G \in \mathbf{R}^{m \times p}$, instead of in $\mathbf{R}^{m \times n}$, and algorithm 1 can be readily applied by adapting the nonlinear mapping $\tilde{f}$ as

$$\tilde{f}_p \colon (\alpha, \beta) \mapsto (\alpha\beta, \ (1-\alpha)^1\alpha\beta, \ \cdots, \ (1-\alpha)^{p-1}\alpha\beta).$$

The aforementioned truncated polynomial approximation on $F$ can be interpreted as follows: For each time step $t$, we approximate the $i$th entry of the value function $x(t)$ as

$$x_i(t) = \sum_{k=1}^t (1-\alpha_i)^{t-k}\alpha_i\beta_i u_i(k) \approx \sum_{k=t-p+1}^t (1-\alpha_i)^{t-k}\alpha_i\beta_i u_i(k), \quad i = 1, \ldots, m,$$

*i.e.*, only the last $p$ reward signal $u(t-p+1), \ldots, u(t)$ (zero padding when $t - p + 1 \le 0$) have an influence on the current value function $x(t)$. As two extreme examples, if $p = n$, we recover the original problem (2.1), where no truncation on history reward signal is applied; if $p = 1$, $x(t)$ can be determined only from $u(t)$, *i.e.*, there is no "memory" in the decision process.

In practice, solving the problem (3.4) can provide a good approximate solution for (2.1) (see §4 for detailed numerical results), with the advantage of very fast computing time. However, we should note that the exact solution condition in §3.2 does not hold anymore for the approximated IMAB problem (3.4), since (3.4) is not equivalent to (2.1) as (2.4) is, although they share exactly the same structure.

## 4 Numerical results

In this section we compare our heuristic for solving (1.1) (including one with truncated polynomial approximation) with direct methods and Monte Carlo methods. We specified and solved the relaxed problem

(2.9), *i.e.*, the step I problem of algorithm 1, using `CVXPY`; the step II problem was solved using `SciPy` with *constrained optimization by linear approximation* (COBYLA) algorithm (Powell, 1994). The number of repeats $N = 10$, and the tolerance $\epsilon = mn\tilde{\epsilon}$, where $\tilde{\epsilon} = 10^{-5}$, *i.e.*, we allow a tolerance of $10^{-5}$ for each entry of the problem data $G^\star$ obtained from the solution of (2.9). The initial values of variables $\alpha_1, \ldots, \alpha_m$ and $\beta_1, \ldots, \beta_m$ were randomly drawn from uniform distributions on $[0, 1]$ and $[0, 5]$, respectively. The direct method was implemented using `SciPy` with the same solver, COBYLA, as in our heuristic. The number of repeats was again set to 10 with initial values of the variables $\alpha_i$ and $\beta_i$ drawn from uniform distributions on $[0, 1]$ and $[0, 5]$, respectively. The Monte Carlo method was implemented using `PyMC`, with the number of samples for tuning the Markov chain and estimating the posterior distribution of the variables set as 2000 and 5000, respectively. The prior for $\alpha_1, \ldots, \alpha_m$ is defined as the uniform distribution on the interval $[0, 1]$; the prior for $\beta_1, \ldots, \beta_m$ is given by a halfnormal distribution with variance $\sigma = 2$. Note that for methods where repeated initialization is required, all computing time is reported as the cumulative time across repeats, *i.e.*, the total time for solving the problem $N$ times. All experiments were performed on an AMD Ryzen™ 9 7950X (4.5 GHz) CPU. The corresponding code to reproduce our results can be found at

https://anonymous.4open.science/r/cvx_imab-A8C0.

### 4.1 The 2-armed bandit testbed

We start from an instance of the forgetting Q-learning model fitting problem (1.7) on a 2-armed bandit environment. It is the most widely considered application scenario of the IMAB problem (1.1) in the computational science field. The demonstrating agent's behavior was implemented according to (1.3) to (1.5). We simulated the agent for 1000 times with each episode consisting of 200 actions, using different values of $\alpha$ and $\beta$ randomly picked from a uniform distribution on $[0, 1]$ and $[0, 5]$, respectively.

Figure 1 shows the comparison between the best estimated log-likelihood value from only solving the relaxed problem of (1.7), *i.e.*, the optimal objective value of the step I problem (2.9), and the log-likelihood obtained by substituting the true parameters $\alpha$ and $\beta$, for the 1000 simulations. The data points align well with the diagonal (the dashed line), but with a slightly upward shift. Such result supports our claim in the last part of §2.2 that solving the relaxed problem (2.9) provides a lower bound to the original nonconvex IMAB problem.

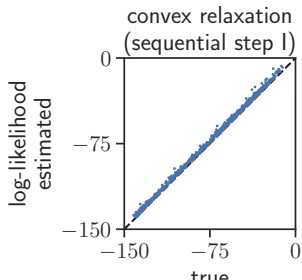

Figure 1: True and estimated log-likelihood from solving (2.9) in the 2-armed bandit testbed.

Figure 2 shows the comparison between our sequential heuristic and the other two different methods. First of all, from the log-likelihood result (top row), we observe that the Monte Carlo method has the best performance, since all the data points are on the diagonal. The sequential heuristic without approximation has the second best performance with few off-diagonal data points, and the direct method performs the worst in this task. Such observation is consistent with the comparison between the estimated and true parameters $\alpha$, $\beta$ (two rows in the middle of figure 2). The parameters estimated from Monte Carlo methods align best with the ground truth, whereas the direct method tends to underestimate $\alpha$ and $\beta$. Together with the computing time for these methods (figure 2, bottom right), our sequential heuristic method achieves almost the same level of performance as the Monte Carlo methods, but with approximately $1/10$ of computing time. The computing time of the direct method are at the same level as our heuristic (*i.e.*, several hundred milliseconds), but the performance is significantly worse. As expected, by introducing a $p = 5$ truncated polynomial approximation to our heuristic, the performance decrease as a result of approximating the matrix $F \in \mathbf{R}^{2 \times 200}$ with $F_p \in \mathbf{R}^{2 \times 5}$ is only minor, while the gain in spared computing time is about five times.

The suboptimality of the obtained solution from our sequential heuristic (without approximation on $F$) can be evaluated according to the gap between the upper and lower bound of the optimal value of (1.7), which is readily obtained as byproduct of algorithm 1. Specifically, solving the relaxed problem (2.9) provides a lower bound $J^{\text{lb}}$ of the objective $J$ in (1.7), and the upper bound $J^{\text{ub}}$ corresponds to the objective function $J$ evaluated with the sequential heuristic solution. The bottom left histogram in figure 2 indicates that the gap

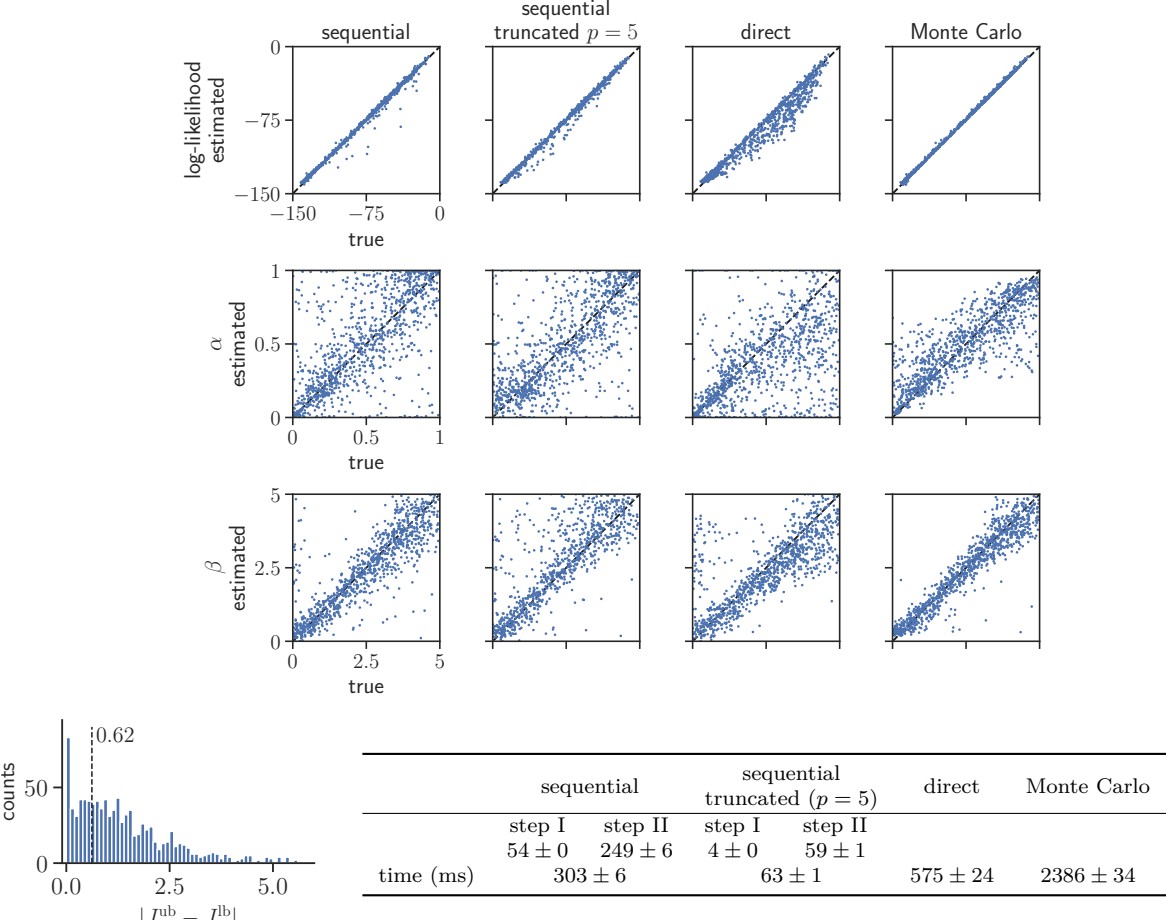

Figure 2: True and estimated log-likelihood (top row) and parameters $\alpha, \beta$ (two middle rows) obtained via different methods in the 2-armed bandit testbed. The bottom left figure shows the histogram of the gap $|J^{\text{ub}} - J^{\text{lb}}|$ from the sequential heuristic. The dashed line corresponds to the largest objective gap across the episodes to which we assigned a certificate for exact solution. The bottom right table shows the computing time (mean $\pm$ standard error across the 1000 episodes) for different methods.

$|J^{\text{ub}} - J^{\text{lb}}|$ for the 1000 episodes are mostly below 5, and specifically, the maximum objective gap within the episodes on which global optimum was achieved by our heuristic method is 0.62. Note that this histogram provides similar information as in the first row of figure 2, but can always be obtained from the solving process even when the true log-likelihood value is unknown (which is the most common case in practice).

## 4.2 The 10-armed bandit with subreward signals

We now consider a more complex environment consists of a 10-armed bandit with subreward signals $u^{\text{rew}}(t), u^{\text{act}}(t) \in \mathbf{R}^{10}$ given by

$$u_i^{\text{rew}}(t) = \begin{cases} 1 & \text{if action } i \text{ was selected } and \text{ rewarded} \\ 0 & \text{otherwise,} \end{cases}$$

and

$$u_i^{\text{act}}(t) = \begin{cases} 1 & \text{if action } i \text{ was selected} \\ 0 & \text{otherwise.} \end{cases}$$

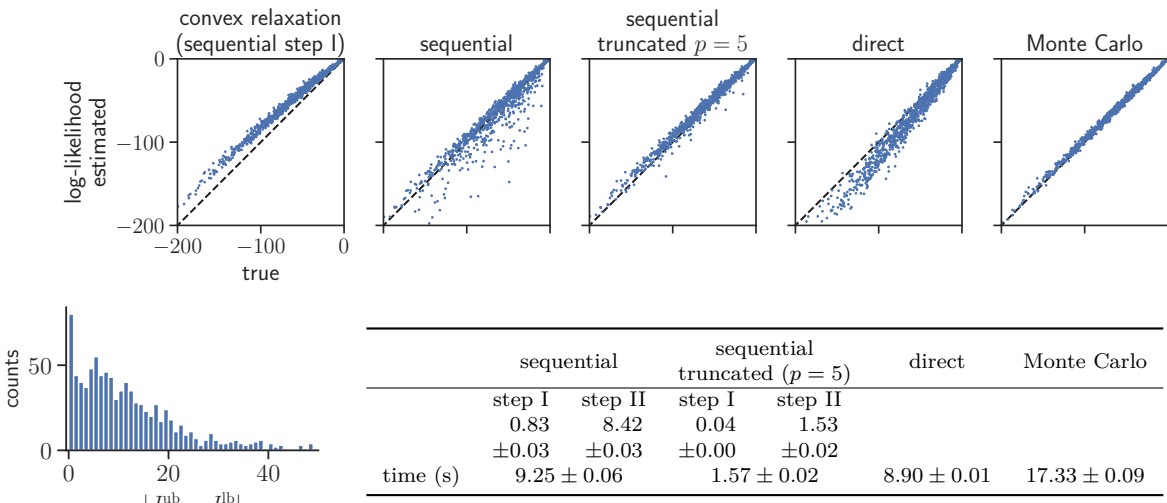

Figure 3: True and estimated log-likelihood (top) obtained via different methods in the 10-armed bandit testbed. The bottom left figure shows the histogram of the gap $|J^{\text{ub}} - J^{\text{lb}}|$ from the sequential heuristic. The bottom right table shows the computing time (mean $\pm$ standard error across the 1000 episodes) for different methods.

The subvalue functions $z^{\text{rew}}(t), z^{\text{act}}(t) \in \mathbf{R}^{10}$ are updated according to

$$z^{\text{rew}}(t) = A^{\text{rew}} z^{\text{rew}}(t-1) + B^{\text{rew}} u^{\text{rew}}(t), \quad z^{\text{act}}(t) = A^{\text{act}} z^{\text{act}}(t-1) + B^{\text{act}} u^{\text{act}}(t),$$

where

$$A^{\text{rew}} = I - \mathbf{diag}(\alpha_1^{\text{rew}}, \dots, \alpha_{10}^{\text{rew}}), \quad B^{\text{rew}} = \mathbf{diag}(\alpha_1^{\text{rew}}\beta_1^{\text{rew}}, \dots, \alpha_{10}^{\text{rew}}\beta_{10}^{\text{rew}}),$$
$$A^{\text{act}} = I - \mathbf{diag}(\alpha_1^{\text{act}}, \dots, \alpha_{10}^{\text{act}}), \quad B^{\text{act}} = \mathbf{diag}(\alpha_1^{\text{act}}\beta_1^{\text{act}}, \dots, \alpha_{10}^{\text{act}}\beta_{10}^{\text{act}}).$$

The reward function $x(t) \in \mathbf{R}^{10}$ used for action selection (1.5) is given by

$$x(t) = \begin{bmatrix} z^{\text{rew}}(t) & z^{\text{act}}(t) \end{bmatrix} \mathbf{1} = z^{\text{rew}}(t) + z^{\text{act}}(t).$$

The IMAB problem under this setup is an instance of (1.8) (which is a larger scale version of the problem combining Beron et al. (2022) and Hattori et al. (2019; 2023)) with variables $\alpha_i^{\text{rew}}, \beta_i^{\text{rew}}, \alpha_i^{\text{act}}, \beta_i^{\text{act}}$, $i = 1, \dots, 10$. We simulated the agent for 1000 times with each episode consisting of 200 actions, using different values of $\alpha_i^{\text{rew}}$, $\beta_i^{\text{rew}}$, $\alpha_i^{\text{act}}$, and $\beta_i^{\text{act}}$ randomly picked from a uniform distribution on $[0, 1]$, $[0, 10]$, $[0, 1]$, and $[0, 5]$, respectively.

Figure 3 shows the comparison between the true and estimated log-likelihood from different methods. Similar to our first experiment, solving the relaxed problem of IMAB returned a lower bound to the optimal value of (1.8) (figure 3, first column). The Monte Carlo method again contributes to the best performance with longest running time. Our sequential heuristic uses a running time at the same level with the direct method, but has significantly better performance. Surprisingly, introducing the truncated polynomial approximation with $p = 5$ not only leads to the fastest computation, but also better performance than the heuristic without approximation. Our sequential heuristic method does not achieve global optimality for all 1000 episodes in this experiment under the given tolerance $\epsilon$, but we can nevertheless obtain some information about the suboptimality of the approximated solutions, as shown in the bottom left histogram of figure 3.

The worse performance of the sequential heuristic without incorporating approximation on $F$ might be explained by the cumulated numerical imprecision. When we solve the step II problem for the full matrix $F$ given data $(G^{\text{rew}})^\star, (G^{\text{act}})^\star \in \mathbf{R}^{10 \times 200}$, we take the fitting residuals at the tails of each row of $(G^{\text{rew}})^\star$ and $(G^{\text{act}})^\star$ into the calculation of the objective $L$. However, the matrices $(G^{\text{rew}})^\star$ and $(G^{\text{act}})^\star$ are obtained from solving the convex problem in step I, where the tails of each row are expected to be very close to zero and thus can be less precisely estimated. It then suggests that the fitting result of step II could be strongly

biased by the noninformative noisy tails. Such an influence is only minor when we solve the step II problem for the truncated matrix $F_p$, with data $(G^{\mathrm{rew}})^\star, (G^{\mathrm{act}})^\star \in \mathbf{R}^{10 \times 5}$, since in this case the matrices $(G^{\mathrm{rew}})^\star$ and $(G^{\mathrm{act}})^\star$ are expected to only have nonzero entries (analogous to only keeping the leading entries of each row in the former case for the step II problem), and thus the influence of the imprecise tails is avoided.

## 5 Discussion and comments

Combining the aforementioned theoretical and numerical results, our sequential heuristic, algorithm 1, for solving the IMAB problem has the following advantages: Firstly, our method is able to achieve similar performance as the Monte Carlo method, at a similar level of time cost as the direct method. The computing time can be further shortened by incorporating the truncated polynomial approximation, at a cost of slightly losing performance. Such an approximation can sometimes be numerically more stable than the original procedure. Moreover, our method enables the evaluation of the suboptimality of the obtained approximate solution to (1.1), via the gap between the upper and lower bound of the objective function $J$. Specifically, this value (or the gap between the upper and lower bound of the objective $L$ of (3.1)) also serves as a certificate for whether our heuristic has achieved global optimum of (1.1). As for direct methods, the strongest claim we could make about the result's suboptimality is that the obtained solution is the best across the $N$ repetitions, even if the best solution does achieve the global optimum. Such feature of our method can be valuable for computational science applications.

Solving the relaxed IMAB problem (2.9) alone, *i.e.*, quit the algorithm 1 directly after step I, can also provide useful information in practice. For example, if the users are only interested in estimating the optimal value functions $x^\star(t)$ for $t = 1, \ldots, n$ of an IMAB problem, instead of recovering the full group of parameters $A$ and $B$, solving the relaxed convex problem (2.9) can provide a robust approximation of the optimal $x^\star(t)$ corresponding to the original problem (1.1) in very short computing time. Besides, although it is not mentioned explicitly in this paper, the lower bound of the optimal value of (1.1) obtained from solving (2.9) is universal, *i.e.*, can be applied to evaluate the solution from any IMAB problem solver. For instance, if one (for some reason) uses a direct solver for the IMAB problem but would like to tune the number of repeated initializations, it would be helpful to first run the step I procedure with the given problem data to obtain a lower bound of (1.1). Then by calculating the objective gap evaluated for the best performed variables from the direct solver across the current number of repetitions, the user can decide whether the number of repeats needs to be increased to further decrease the gap.

Note that in both experiments of §4, the computing time for the step II procedure were much longer than that for step I. On the one hand, this is because we have to solve the optimization problem repeatedly during step II. On the other hand, the selected solver, COBYLA, is some variants of the subgradient methods (Shor, 1985), which is not the fastest among the other supported solvers for local minimization problems with constraints, but is numerically more stable in our experiment setup. In practice, one can select a different solver for step II according to the application scenario to save computing time, for example, incorporating the second-order derivative information using sequential quadratic programming (Kraft, 1988), or allowing GPU acceleration using `JAX`, just list a few.

The analysis and convexification of the IMAB problem discussed in §2 builds upon and extends the work by Beron et al. (2022). In their work, they provides discussion about the (approximate) equivalence relationship between the forgetting Q-learning model and a logistic regression model, from the behavior modelling point of view, and incorporating some experimental observations. We extend their discussion to general IMAB problems and provide some theoretical results from the convex analysis perspective.

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

# A    The step II problem in convex form

In this section, we discuss an option of selecting the penalty function $\phi$ in step II of algorithm 1, such that the corresponding optimization problem is convex. Recall that the problem (3.1) has the general form

$$
\begin{array}{ll}
\text{minimize} & L = \sum_{i=1}^{m} \phi(\tilde{f}(\alpha_i, \beta_i), g_i^{\star}) \\
\text{subject to} & 0 \le \alpha_i \le 1, \quad \beta_i \ge 0, \quad i = 1, \ldots, m
\end{array}
\tag{A.1}
$$

with variables $\alpha_i, \beta_i \in \mathbf{R}$, $i = 1, \ldots, m$ and data $g_i^{\star} \in \mathbf{R}^n$, $i = 1, \ldots, m$, where the nonlinear transformation $\tilde{f}$ is given by

$$
\tilde{f} \colon (\alpha, \beta) \mapsto (\alpha\beta,\ (1-\alpha)^1 \alpha\beta,\ \cdots,\ (1-\alpha)^{n-1}\alpha\beta).
$$

Consider the least squares penalty function in the log space[2], *i.e.*,

$$
\phi(u, v) = \|\log u - \log v\|_2^2.
\tag{A.2}
$$

The objective function $L$ of (A.1) is then

$$
\begin{aligned}
L &= \sum_{i=1}^{m} \phi(\tilde{f}(\alpha_i, \beta_i), g_i^{\star}) = \sum_{i=1}^{m} \|\log \tilde{f}(\alpha_i, \beta_i) - \log g_i^{\star}\|_2^2 \\
&= \sum_{i=1}^{m} \left\| \begin{bmatrix} \log(\alpha_i\beta_i) \\ \log((1-\alpha_i)^1 \alpha_i\beta_i) \\ \vdots \\ \log((1-\alpha_i)^{n-1} \alpha_i\beta_i) \end{bmatrix} - \log g_i^{\star} \right\|_2^2 \\
&= \sum_{i=1}^{m} \left\| \begin{bmatrix} 0 \times \log(1-\alpha_i) + \log(\alpha_i\beta_i) \\ 1 \times \log(1-\alpha_i) + \log(\alpha_i\beta_i) \\ \vdots \\ (n-1) \times \log(1-\alpha_i) + \log(\alpha_i\beta_i) \end{bmatrix} - \log g_i^{\star} \right\|_2^2.
\end{aligned}
$$

Introducing the variable transformations:

$$
A = \begin{bmatrix} 0 & 1 \\ 1 & 1 \\ \vdots & \vdots \\ n-1 & 1 \end{bmatrix}, \quad x_i = \begin{bmatrix} \log(1-\alpha_i) \\ \log(\alpha_i\beta_i) \end{bmatrix}, \quad b_i = \log g_i^{\star},
$$

we transform the step II problem (A.1) with penalty function (A.2) into an equivalent constrained least squares problem

$$
\begin{array}{ll}
\text{minimize} & \sum_{i=1}^{m} \|Ax_i - b_i\|_2^2 \\
\text{subject to} & (x_i)_1 < 0, \quad i = 1, \ldots, m
\end{array}
\tag{A.3}
$$

with variables $x_i \in \mathbf{R}^2$ and data $A \in \mathbf{R}^{n \times 2}$, $b_i \in \mathbf{R}^n$. (The notation $(x_i)_1$ denotes the first entry of vector $x_i$.)

The advantage of considering the least squares penalty in the log space, given by $\phi(u, v) = \|\log u - \log v\|_2^2$, for the problem (A.1) is that the step II problem (A.3) is now convex, such that repeated initializations are avoided. However, the major drawback of (A.3) is that the penalty function (A.2) tends to add very large penalty to those entries of $g_i^{\star}$ that are small, compared to those entries that are away from zero, even if they have the same residual. Such behavior is due to the logarithm function $u \mapsto \log u$, whose slope approaches infinity as $u \to 0$. Recall that in (2.9) we expect the entries of vector $(g_i^{\star})_j$, $j = 1, \ldots, n$ to decay (ideally exponentially) as the index $j$ increases. Hence, the solution of (A.3) using data $g_1^{\star}, \ldots, g_m^{\star}$ from the solution

---

[2]By considering such a constraint we implicitly require that the strict inequalities $0 < \alpha_i < 1$ and $\beta_i > 0$ hold for $i = 1, \ldots, m$ (which do hold for almost all IMAB applications in practice).

of (2.9) tends to have a very good fit to the tail of the vectors $g_1^\star, \ldots, g_m^\star$ where the entries are nearly zero, while the residual at the beginning entries are less irritative and thus can be very large. If $n$ is small, or one considers solving the truncated problem (3.4) with small $p$, taking (A.3) as the step II of algorithm 1 can provide a stable optimal solution; if $n$ is large, *i.e.*, there might exist lots of nearly zero entries in $g_i^\star$, the results can be problematic since it tends to provide a fitting that matches the noninformative small tails of $g_i^\star$ which are largely influenced by numerical roundoff error, instead of the informative entries at the beginning. For this reason, although we demonstrate this option of formulating the step II problem as the convex problem (A.3), we will not consider this approach further in this work.

