# OpenReview forum: "Solving Inverse Problem for Multi-armed Bandits via Convex Optimization"
_TMLR — Withdrawn by Authors_

### Review · Reviewer_SYe6 · 2025-02-14

**Summary Of Contributions:**

This work introduces a heuristic methodology for addressing the inverse (identifying/estimating the parameters) multi-armed bandit (linear control) problem (IMAB), formally defined in Equations (1.1), (1.7), and (1.8). The authors motivate this framework by drawing connections to applications in neuroscience and psychology, specifically within the context of "forgetting Q-learning model fitting."

Methodologically, the paper characterizes the non-convex nature of the IMAB problem and proposes a relaxation to a convex formulation. This relaxation is achieved through a variable transformation, enabling efficient optimization. The solution obtained from this convex relaxation is subsequently mapped back to the original problem space via an $L_2$ minimization.

The proposed approach consists of a two-step sequential heuristic:

1. The original constraints, which enforce exponential decay along the rows of the transformed matrix, are relaxed to require only decay along each row, without the exponential constraint. This relaxed problem is convex and efficiently solvable.

2. A set of variables satisfying the original formulation is sought, by minimizing the $L_2$ distance to the solution obtained in Step 1. This step involves a non-convex optimization. The authors justify their approach of avoiding convexification and least-squares minimization in log-space by highlighting potential issues with the behavior of such methods with respect to small solution values.

The authors claim that their two-step method provides ``certificates'' indicating whether the solution is exact or approximate.

The experimental evaluation compares the proposed method against a Monte Carlo-based inverse solution and an existing state-of-the-art direct method.  The results demonstrate the computational advantages of the proposed approach.

**Audience:**

No

**Claims And Evidence:**

No

**Requested Changes:**

The following adjustments are proposed to strengthen and enhance the manuscript:

- **Motivation and Justification (Critical)**:  Beyond the demonstrated reduction in computational complexity, a more compelling case for the benefits of the proposed heuristic would be appreciated.
  - Specifically, a more in-depth discussion on whether the proposed approach offers significant advantages over Monte Carlo methods, is crucial.
  - A more detailed and nuanced discussion of the trade-offs between computational cost and solution accuracy would be of help.

- **Connections to Control Literature (Strengthening)**: Exploring the connections between the problem formulation presented in this work and related concepts in the control literature would be a valuable addition.  Specifically, the relationship between Equation (1.1) and linear control problems, particularly in the context of system identification (parameter estimation), should be discussed.  This would enrich the related work section and provide valuable context for readers familiar with control theory.

- **Optimality Certificates (Critical)**:  The claim in Section 3.2 regarding the "exact" solution obtained under certain conditions (early stopping with  $|𝐿^{(𝑘)}−𝐿^{(lb)} \leq \epsilon$) requires clarification.
  - To the best of my understanding, Equation (3.2) holds only within a tolerance of $\epsilon$.
  - Therefore, is the solution obtained truly exact, or is it $\epsilon$-close to the optimal solution for Equation (2.1)?
  - A precise definition and rigorous justification of these certificates are essential.

- **Evaluation Metrics (Critical)**: While the experiments focus on log-likelihood estimation, the core problem formulation aims to recover the ground-truth parameters $\alpha$ and $\beta$.
  - The evaluation should include metrics assessing the accuracy of the estimated parameters compared to their optimal counterparts.
  - Quantifying the closeness of the computed solutions ($F^*$ and $G^*$) to the true counterparts would also help demonstrate the effectiveness of the proposed method.

- **Computational Performance Analysis (Critical)**:  It is unclear whether the reported computational differences are solely attributable to algorithmic complexity or are influenced by specific implementation details. Hence, I encourage a more rigorous analysis.
  - For instance, consider fixing the computation time for given implementations and comparing the solution accuracy achieved within that time limit.

- **Experimental Setup and Parameter Initialization (Strengthening)**: The discrepancy between the parameter distributions used for data generation (uniform distributions on [0,1] and [0,5]) and the initialization of the Monte Carlo method (half-normal distribution with variance $\sigma = 2$) needs to be addressed.
  - Using consistent priors for both data generation and sampling is essential for a fair comparison.
  - Please justify the current setup or revise it to ensure consistency.

- **Typographical Errors and Clarifications (Strengthening)**:
  - In the unnumbered equation for the objective function J (between Equations 2.4 and 2.5), the second equality appears incorrect.
    - Should the first term of the sum be $log (y(t)^\top) x(t)$?
  - In Algorithm 1, Step I, an "exp" might be missing in the relaxed problem formulation.
    - Should it read $\log(\frac{y(t)^{\top} exp(diag(GU(t)))}{(1^{\top} exp(diag(GU(t)))}$?
  - The introduction of Equation (1.1) without the subsequent context provided in Section 1.1 feels somewhat abrupt. Furthermore, the variations of Equation (1.1) presented in Equations (1.7) and (1.8) before the specific instance in Equation (2.1) are also interesting and important.
    - Consider reorganizing the Introduction section to provide a more gradual and accessible introduction to the problem.

**Strengths And Weaknesses:**

This submission presents several strengths:

- **Clarity of Presentation**: The manuscript is well-written, with clear articulation of the research goals, underlying ideas, and proposed approach. (See minor comments below for specific suggestions.)

- **Problem Characterization**: The authors effectively demonstrate the non-convex nature of the IMAB problem, providing valuable insight into its complexity.

- **Computational Efficiency**: The experimental results demonstrate that the proposed methodology achieves comparable performance to Monte Carlo methods, but with a reduced computational cost. This is the main advantage.

However, the submission also presents some areas that require further attention:

- **Significance and Impact**: While the authors acknowledge that Monte Carlo methods offer superior performance, albeit with potential implementation challenges and computational demands, the significance and impact of the proposed method could be strengthened. A more in-depth discussion (see some suggestions below) of the trade-off between performance and computational cost would enhance the paper.

- **Optimality Certificates**: The authors' claim about the provision of certificates for global optimality under certain circumstances requires further clarification. A precise statement of these circumstances and a detailed explanation of how these certificates are derived are essential for a complete understanding of the method's theoretical properties. Providing a more rigorous analysis of the approximation guarantees would significantly strengthen the paper.

---

### Review · Reviewer_e7pG · 2025-03-02

**Summary Of Contributions:**

The manuscript addresses the problem of inverse multi-armed bandits (IMAB), where the objective is to infer model parameters (e.g., learning rates and latent variables for arm draws) from observed actions and rewards. The authors propose a two-stage algorithm that leverages a convex relaxation of the underlying nonconvex problem. Numerical results show that this method can often attain performance comparable to Monte Carlo sampling approaches at a fraction of the computational cost.

**Audience:**

Yes

**Broader Impact Concerns:**

N/A.

**Claims And Evidence:**

Yes

**Requested Changes:**

1. It is somewhat unclear which variables represent observable data and which are optimized. By the end of the draft, I understood that “problem variables” mean the target parameters in the optimization, while “problem data” refer to the observed values. Explicitly stating something like “minimize over \(x(t)\in\mathbb{R}^m, A,\ldots, B,\ldots\), given observed \(y(t)\), etc.” would improve readability. (I come from a statistics background, so perhaps I am just not used to the current phrasing.)

2. On page 1, the authors state: “Specifically, we assume that the matrices \(A\) and \(B\) are diagonal and \(y(t)\)\dots The objective function has the form \(J = \dots\).” However, I suspect the results also rely on the specific form of \(\ell\) and the learning process (as in Equations (1.5)–(1.6)). If so, the page 1 description might be slightly confusing, as the assumptions there alone do not fully imply the stated conclusions.

3. Could the authors give a brief empirical or conceptual illustration of the decision-making process described on pages 1 and 2? Even a short description of potential applications or real-world settings would help readers outside behavioral neuroscience understand why these particular assumptions are relevant.

**Strengths And Weaknesses:**

### Strengths
- Demonstrating how to transform a nonconvex fitting problem into a relaxed convex problem is both original and practically useful.
- The paper carefully explains why IMAB formulations are nonconvex, emphasizing the polynomial dependence of parameters.
- Deriving a convex relaxation by replacing exponential decay constraints with monotonic decay constraints is well-reasoned and provides a lower bound on the original objective.
- The authors show that in certain cases, Algorithm 1 can yield a globally optimal solution, highlighting the advantage of the proposed approach.

### Weaknesses

1. **Problem Setting**
   - While the authors refer to their formulation as a multi-armed bandit (MAB) problem, it seems to be a specific instance in which arms are drawn according to Equation (1.5), alongside other conditions determined by the authors.
   - Personally, I find it difficult to interpret this as a standard MAB problem. Beyond the paper’s technical contributions, the motivation for studying this specific setup is not entirely clear to me. I do not claim that the setting is nonstandard or inappropriate for the community, only that I am unfamiliar with it.
   - Is this problem the inverse RL, rather than the IMAB?

2. **Range of Models**
   - The paper focuses primarily on forgetting Q-learning. If the approach can be extended to a broader class of IMAB formulations, providing additional examples or references would clarify how widely this relaxation technique applies—and under what limitations—across other bandit or RL contexts.

I should note that I am not deeply familiar with behavioral or neuroscience research or the particular decision-making process described by the authors. I mainly work on bandit problems in areas like stochastic or adversarial bandits (e.g., advertising and marketing), which differ substantially in their assumptions and methods. Consequently, I cannot fully judge whether this approach is novel or broadly beneficial for the neuroscience community. While both fields use the term multi-armed bandit, it seems that the frameworks are quite distinct. I therefore believe it would be best evaluated by other reviewers who focus on similar settings.

---

### Review · Reviewer_QR2a · 2025-03-02

**Summary Of Contributions:**

The paper proposes a solution to a non-convex optimisation problem that models an inverse multi-armed bandit problem. The method is elegant: first the problem is reformulated into another constrained problem and relaxed into a convex problem. The resolution of this new relaxed convex problem can only be done partially with numerical tools, so a second part proposes a heuristic to solve a remaining function-fitting non-convex problem. The final solution can be implemented using several numerical tools and experiments are run to illustrate and test the method.

**Audience:**

Yes

**Claims And Evidence:**

Yes

**Requested Changes:**

Technical questions:

* Can you explain again how you get the equality below 2.4? I don’t see how the y(t) get out of the log. The scalar product is a sum, right? What am I missing?
* I was a bit lost reading the experiment: what is your method, what is SOTA? Or are these all “versions” of your method? I felt it was a bit hard to find what is what, maybe it could be a bit more clearly structured, perhaps indicating these names already in section 3?

**Strengths And Weaknesses:**

## Strengths:

* The problem is elegant and its solution is elegant, using several clever ideas, both from convex optimisation theory and from numerics
* The paper is well written and easy to follow
* The inverse MAB is quite a common problem in neuroscience and I’d expect this method to be used where applicable. But I am not an expert.


## Weaknesses :

* I have a couple of relatively minor technical questions, see below.
* Inverse MAB may also refer to recovering the underlying reward function for each arm (that is the probability that u_i(t)=1 conditionally on arm I being pulled) when rewards themselves are not observed but a learning agent is observed [Guo21]. Here, the goal is to be to recover the intrinsic mechanism of the forgetting Q-Learning agent (parameters \alpha_k,\beta_k). Given the risk of confusion, I think it’d be good to have a clear paragraph somewhere at the beginning that says explicitly what is the goal, what are the input and the output(s) of your algorithm.
* The Inverse MAB problem modelled here is a specific instance where rewards are binary. I’m guessing the method could extend to more general (bounded) rewards but maybe it’s worth commenting on this.


[Guo21] Guo, Wenshuo, et al. "Learning from an exploring demonstrator: Optimal reward estimation for bandits." arXiv preprint arXiv:2106.14866 (2021).

---

### Note · Authors · 2025-03-05

**Comment:**

Thanks so much to the action editor and reviewers for their work on this paper. We appreciate the valuable comments and decide to withdraw this paper for further improvement.

**Withdrawal Confirmation:**

I have read and agree with the venue's withdrawal policy on behalf of myself and my co-authors.